# Effects and utility of an online forward triage tool during the SARS-CoV-2 pandemic: a mixed method study and patient perspectives, Switzerland

Janet Michel ,[1] Annette Mettler,[1] Raphael Stuber ,[1] Martin Müller,[1] Meret E Ricklin,[1] Philipp Jent ,[2] Wolf E Hautz ,[1,3] Thomas C Sauter [1]

JM and AM are joint first authors.

¹Department of Emergency Medicine, Inselspital, University Hospital, University of Bern, Bern, Switzerland
²Department of Infectious Diseases, Inselspital, University Hospital, University of Bern, Bern, Switzerland
³Centre for Educational Measurement, University of Oslo, Oslo, Norway

**Correspondence to**
Dr Janet Michel;
janetmichel71@gmail.com

## ABSTRACT

**Objective** To assess the effects (quantitatively) and the utility (qualitatively) of a COVID-19 online forward triage tool (OFTT) in a pandemic context.

**Design** A mixed method sequential explanatory study was employed. Quantitative data of all OFTT users, between 2 March 2020 and 12 May 2020, were collected. Second, qualitative data were collected through key informant interviews (n=19) to explain the quantitative findings, explore tool utility, user experience and elicit recommendations.

**Setting** The working group e-emergency medicine at the emergency department developed an OFTT, which was made available online.

**Participants** Participants included all users above the age of 18 that used the OFTT between 2 March 2020 and 12 May 2020.

**Intervention** An OFTT that displayed the current test recommendations of the Federal Office of Public Health on whether someone needed testing for COVID-19 or not. No diagnosis was provided.

**Results** In the study period, 6272 users consulted our OFTT; 40.2% (1626/4049) would have contacted a healthcare provider had the tool not existed. 560 participants consented to a follow-up survey and provided a valid email address. 31.4% (176/560) participants returned a complete follow-up questionnaire. 84.7% (149/176) followed the recommendations given. 41.5% (73/176) reported that their fear was allayed after using the tool. Qualitatively, seven overarching themes emerged namely (1) accessibility of tool, (2) user-friendliness of tool, (3) utility of tool as an information source, (4) utility of tool in allaying fear and anxiety, (5) utility of tool in medical decision-making (6) utility of tool in reducing the potential for onward transmissions and (7) utility of tool in reducing health system burden.

**Conclusion** Our findings demonstrated that a COVID-19 OFTT does not only reduce the health system burden but can also serve as an information source, reduce anxiety and fear, reduce potential for cross infections and facilitate medical decision-making.

## INTRODUCTION

The number of COVID-19 cases across the globe has surpassed 25 million and incident rates are again on the rise as many European countries experience subsequent waves.[1–4] Many people are seeking reliable information, recommendations on testing and management of COVID-19 as well as reassurance, adding to the health system burden. Online forward triage tools (OFTTs) are being widely used during this COVID-19 pandemic context[5–8] as misinformation and worry in the population abound. There is evidence from an earlier 2009 H1N1 influenza pandemic that online tools are effective and practical in reducing the health system burden.[9 10] There is also emerging evidence of this nature from the COVID-19 context.[6 11–14] For example, OFTTs help reduce exposure of worried but uninfected and infected persons, through avoidance of hospitals and doctors' offices—enabling patients to access recommendations of what to do, from the comfort of their own homes.[10 11]

Using OFTTs is relatively easy to the computer-literate. People respond to questions and on completion, recommendations are given, for example, isolate, test, do not test etc. Existing evidence on the effects and utility of OFTTs differ with possible implications on the quality of the symptom assessment.[5] According to the literature, the reasons patients use symptom checkers or

OFTTs are (1) to understand the causes of their symptoms (76%), (2) to determine whether or not to seek care (33%) and (3) where to seek care (21%).[15] There is also evidence that patients who have previously experienced a diagnostic error are more likely to use OFTT to search for where to seek care[15] than those who have not.

### Challenges with OFTT use and research gap

In the European Union, 87% of people aged 75 years and above have never been online according to a recent survey.[16] That means, the elderly may be less inclined to use online tools if not computer-literate. This in turn shuts the elderly out from society, increasing isolation and loneliness, not to mention the missed health benefits.[10] The digital divide is real.[17] How can digital tools be designed to be more inclusive?[18] Information on factors influencing the use of OFTTs is scant and the validation of COVID-19 OFTTs, such as other OFTTs, seems neglected.[15 19] That makes the quality assessment of these tools paramount,[5] as evidence on effects and utility of OFTTs is limited.

### The aim of this study

This study aimed at assessing the effects (quantitatively) and the utility (qualitatively) of a COVID-19 OFTT during a pandemic context in Switzerland, exploring patient perspectives and derive recommendations for tool improvement. We hypothesised that an OFTT adequately reduces patient visits to the healthcare system and consequently reduces the health system burden. We further explored qualitatively, for emergent themes, capturing the tool utility to this population.

## METHODS

### Study design and participants

We employed a mixed method sequential explanatory design to study the utility of the OFTT and the effects of using such a tool. The rationale for mixing both kinds of data within one study is that neither qualitative nor quantitative methods are sufficient by themselves to capture details of a phenomenon. In combination, they complement each other, taking advantage of the strengths of each. As in sequential explanatory designs, quantitative data collection was done first, as a major component of our study to inform qualitative interviews, see figure 1.

### OFTT description and setting

The working group e-emergency medicine at the emergency department, Inselspital, University Hospital Bern, together with the Department of Infectious Diseases, Inselspital, University Hospital Bern, developed an OFTT, which was made available online (coronatest.ch). To the best of our knowledge, this was one of the first COVID-19 OFFTs set up in the German speaking part of Switzerland. In a skip logic, the OFTT displayed the current test recommendations of the Federal Office of Public Health (FOPH) on whether someone needed testing

| Phase | Procedure | Product |
|---|---|---|
| Collection of clinical data | OFTT | Data about usage of the tool Contact data for later study phases |
| Quantitative data collection | Questionnaires | Numeric data |
| Quantitative data analysis | Statistical analysis | Descriptive statistics |
| Connecting quantitative and qualitative data | Purposefully selecting patients for interviews | Interview guide |
| Qualitative data collection | Individual in-depth interviews | Interview transcripts |
| Qualitative data analysis | Coding and thematic analysis | Codes and themes |
| Integration of qualitative and quantitative results | Interpretation and explanation of quantitative results with the aid of and qualitative findings | Implications for OFTT development |

**Figure 1** Mixed methods sequential explanatory study design. OFTT, online forward triage tool.

for COVID-19 or not. No diagnosis was provided by the OFTT.

The questions and the content of the OFTT represented the official FOPH recommendations at the time. Thus, the OFTT was comparable in content to other OFTTs in Switzerland, which were based on the FOPH guidelines within that time period. One additional non-mandatory question, which did not affect the result, was integrated in our OFTT from 11 March 2020, namely the question 'What would you do if this online test did not exist?'.

There were two possible outcomes of the OFTT: 'according to the criteria of the FOPH (BAG), one meets or does not meet the criteria for a test for an infection with the COVID-19'. The results page was linked to the FOPH's official behavioural recommendations and recommendations for the testing process. The average time to complete the assessment was 75 s.

### OFTT triage

Details on the structure of the OFTT as well as screen shot are published in a separate quantitative paper.[20] The FOPH national COVID-19 Swiss testing criteria were transferred into a digital decision tree and adjusted promptly after the criteria were adapted by the FOPH. During the first phase of the pandemic, the recommendations for testing or not testing were mainly based on contact with an infected person or a visit to a risk area and were then changed during the course of the pandemic to a testing regime based on risk groups (healthcare professionals, patients>65 years and patients with pre-existing conditions). With the general availability of the tests, the test recommendations were extended to all symptomatic patients and our OFTT became obsolete. Unlike other triage techniques performed on emergency patients, the aim of the OFTT was not to make a COVID-19 diagnosis, assess the risk of severe COVID-19 progression or recommend treatment (see figure 2 below).

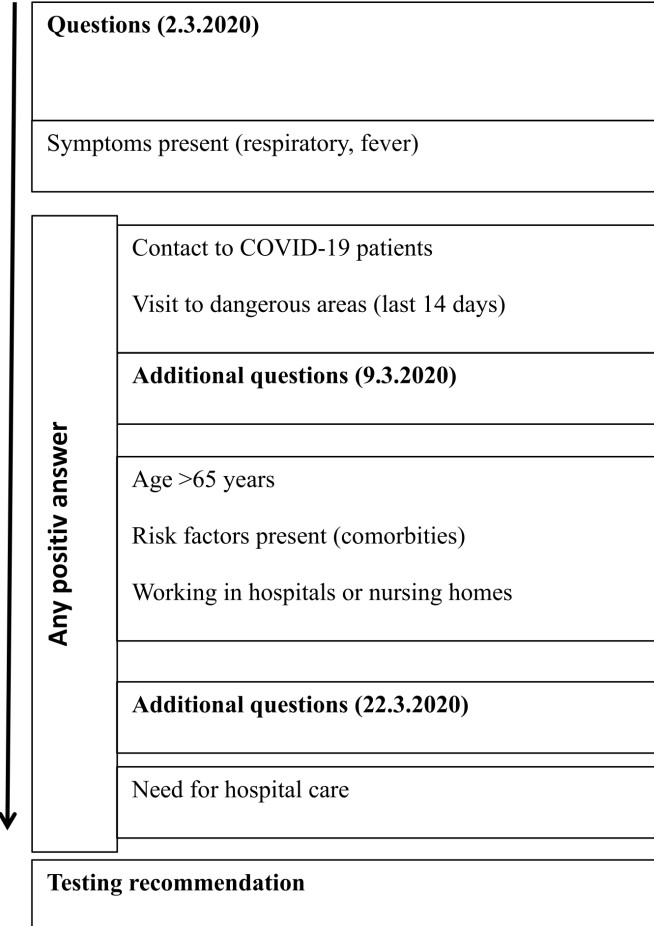

**Questions (2.3.2020)**

Symptoms present (respiratory, fever)

Contact to COVID-19 patients

Visit to dangerous areas (last 14 days)

**Additional questions (9.3.2020)**

Age >65 years

Risk factors present (comorbities)

Working in hospitals or nursing homes

**Additional questions (22.3.2020)**

Need for hospital care

**Testing recommendation**

*Any positiv answer*

**Figure 2** Online forward triage tool triage.

## Quantitative data

### Research participants and data collection

Participants included all users above the age of 18 that used the OFTT between 2 March 2020 and 12 May 2020. In this timeframe, the recommendations on COVID-19 frequently changed in Switzerland and there was an initial lack of testing reagents and capacity as well as the risk of overburdening the healthcare system. During the first few weeks of the pandemic, the FOPH recommended testing only for symptomatic patients after travel to high-risk countries (eg, Italy and China) or symptomatic contacts of patients with COVID-19. In weeks that followed (as from 20 March 2020), the strategy changed to testing of high-risk groups (older than 65 years, pre-existing conditions and healthcare workers). The countries and risk groups were regularly adjusted according to the spread of the virus and the findings about risk groups but also the availability of testing capacity.

Due to the rapid spread of the virus in Switzerland, and broadly available testing capacities, a universal test recommendation was made by the FOPH—on 27 April 2020. All symptomatic individuals were eligible to test. With this recommendation, our OFTT provided less benefit to the user and was finally removed on 12 May 2020 from the website paving the way to a second-generation OFTT.

To minimise the barrier to the use of the OFTT and for legal data protection reasons, no personal data were collected within the OFTT. Further data on the users of the OFTT were collected in a second step, from participants who gave their explicit consent and provided their email addresses to be contacted. This also made it possible to investigate the adherence to recommendations and the test results. A non-mandatory additional question was built into the OFFT from 11 March 2020.

A pretested online questionnaire (see online supplemental file 1) was used to assess the:
1. Utilisation of the OFTT, including way of referral to the tool, reasons for use and information searched.
2. Additional factors, including influence of the media and influence of the OFTT on fear and anxiety.

The database used is compliant with Swiss laws on the collection of personal health-related information. The follow-up questionnaire is available as online supplemental file 1. Due to ethical reasons, we included the option 'not want to answer' as a choice in the questionnaire for the sociodemographic data, in case the respondent did not want to give a statement on this specific sensitive topic.

The qualitative interviews were conducted with purposefully selected key informants who gave their consent during the survey (see below).

### Data analysis

Quantitative data were analysed in Stata V.16.1 (StataCorp, The College Station, Texas, USA). Descriptive statistics for all variables as mean and SD or frequency as determined by the type and distribution of the data were computed. Categorical variables between two groups were compared using $\chi^2$ statistics and the distribution of continuous variables were compared using Wilcoxon rank sum test.

To assess the risk of selection bias and to estimate the similarity of the groups, we compared responses to overlapping questions within the OFTT and the follow-up survey.

## Qualitative data

To explain the quantitative results, we explored the experience of tool use by the patients qualitatively. Following quantitative data analysis, an interview guide was created and adapted iteratively.

### Purposeful and quota sampling

We purposefully sampled participants from those who had first, used our OFTT, second, had taken part in the follow-up survey and third, had consented to a follow-up interview. We included participants of all age groups (quota) to ensure inclusiveness.

### Sample size

Many experts suggest saturation as central to qualitative sampling.[20] In this study, we aimed for both data saturation and rich and detailed narratives and achieved this with 19 key informants from all age groups (see table 1).

## Data collection

Due to COVID-19 concerns, video rather than face-to-face interviews were held with most participants in September 2020. A combination of video and telephonic interviews were conducted with three participants who had technical challenges and a telephone-only interview was held with one woman, aged above 65 years, who had no computer access. Three face-to-face interviews were held with three key informants: one who was a hospital healthcare worker, and two key informants who worked close to Bern University Hospital. A semistructured interview guide informed by the quantitative results was used (see online supplemental file 2). This was adapted iteratively throughout the data collection period. Two qualitative researchers sat in each session fielding questions in turns. All interviews were conducted in German by two researchers fluent in both English and German. The interviews lasted between 45 min to one and a half hours. Two audio recorders were used in each session. All participants gave individual written consent as well as oral consent to the recording at the beginning of each session. See table 1 for summary of key informants.

## Data analysis

Audio recordings were transcribed, analysed and triangulated with quantitative data results. Qualitative narratives were obtained to explain quantitative results as well as to explore the utility of OFTT to patients as well as elicit recommendations to make online tools more useful and inclusive. A grounded theory approach was used. Concepts were identified from collected data and compared iteratively. These concepts were grouped into categories and culminated into the identified themes.

## Measures to ensure trustworthiness of data

To ensure dependability, data collection and analysis were performed iteratively, continuously adjusting our interview guide to capture newly emerging themes. Throughout data collection, two qualitative researchers kept reflexive journals and debriefed at the end of each interview. To ensure transferability, a thick description of participants, context and data collection process has been outlined. Data were managed and analysed with the aid of MAXQDA2018.

| Table 1 | Key informant summary | | |
|---|---|---|---|
| Age group | Males | Females | Total |
| 18–29 | 1 | 2 | 3 |
| 30–45 | 2 | 2 | 4 |
| 46–64 | 3 | 4 | 7 |
| 65+ | 4 | 1 | 5 |
| Total | 10 | 9 | 19 |

## Patients and public involvement statement

Patients and public were not involved in the design, conduct, reporting or dissemination of this research since the OFTT was set up as an emergency response to the pandemic.

## RESULTS
### Quantitative results

In total, n=6272 completed assessments of the OFTT were recorded on the website during the study period from 2 March 2020 to 12 May 2020. This question asked OFTT users what they would have done had the OFTT not existed. The question was answered by 97.6% (3953/4049) of the users as follows: 40.2% (1626/4049) would have contacted the general practitioner (GP) or visited a hospital had the tool not existed; furthermore, 16.4% (665/4049) would have contacted a hotline.

In the OFTT, 25.6% (1608/6272) of assessments received a recommendation to test for COVID-19 during the study period. In the follow-up survey question, 'Did the online tool recommend you to test for COVID-19?'— 31.8% (56/176) answered, yes.

In the OFTT, 13.2% (564/4270) of OFTT users reported being over 65 years of age. The variable age was only included and mandatory during some phases of the study period in accordance with the FOPH guidelines that changed frequently. This resulted in 4270 assessments with data on age. In the follow-up survey, 17.6% (31/176) reported being over 65 years.

A link to the online follow-up questionnaire was sent to 560 participants who consented to a follow-up survey by providing a valid email address. The online questionnaire was filled out by 37.9% (212/560) of the participants; 31.4% (176/560) completed the whole questionnaire and were included in the analysis (all 22 questions, see online supplemental file 1). An overview of sociodemographic characteristics of participants of the follow-up survey are presented in table 2.

The survey revealed that 84.7% (149/176) followed the tool recommendations and stayed at home, thereby reducing the workload of GPs and hospitals. Information about the utilisation of the OFTT, specifically which information was searched for, how subjects found the tool and information about satisfaction with the tool, is presented in table 3.

We present additional factors that may have influenced how individuals coped during the COVID-19 pandemic, their use of the OFTT and adherence to OFTT recommendations. Overarching topics that were asked included the influence of the media, fear and uncertainty, and reasons for adherence to the recommendation (see table 4). All questions and answers from the follow-up questionnaire are attached (see online supplemental file 1).

### Qualitative findings

Seven overarching themes on the utility of the OFTT emerged during the qualitative interviews. These are used

**Table 2** Sociodemographic table of participants of follow-up survey

|  | Total (n=176) | Female (n=101) | Male (n=75) | P value* |
|---|---|---|---|---|
| Age (mean, SD) | 50.1 (±15.4) | 45.9 (±14.1) | 55.7 (±15.4) | <0.001 |
| Education |  |  |  |  |
| Not want to answer | 6 (3.4) | 3 (3.0) | 3 (4.0) |  |
| University | 120 (68.2) | 67 (66.3) | 53 (70.7) |  |
| Higher secondary school | 27 (15.3) | 17 (16.8) | 10 (13.3) |  |
| Lower secondary school | 23 (13.1) | 14 (13.9) | 9 (12.0) | 0.871 |
| Income per month |  |  |  |  |
| Not want to answer | 29 (16.5) | 17 (16.8) | 12 (16.0) |  |
| <SFr4000 | 26 (14.8) | 20 (19.8) | 6 (8.0) |  |
| 4000–6000 | 42 (23.9) | 27 (26.7) | 15 (20.0) |  |
| >6000 | 79 (44.9) | 37 (36.6) | 42 (56.0) | 0.037 |
| Work |  |  |  |  |
| Not want to answer | 33 (18.8) | 14 (13.9) | 19 (25.3) |  |
| Employed | 106 (60.2) | 64 (63.4) | 42 (56.0) |  |
| Self-employed | 24 (13.6) | 13 (12.9) | 11 (14.7) |  |
| Unemployed | 3 (1.7) | 3 (3.0) | 0 (0.0) |  |
| Lost work (COVID-19) | 1 (0.6) | 1 (1.0) | 0 (0.0) |  |
| Student/trainee | 9 (5.1) | 6 (5.9) | 3 (4.0) | 0.236 |
| Insurance |  |  |  |  |
| Do not know | 5 (2.8) | 3 (3.0) | 2 (2.7) |  |
| General | 68 (38.6) | 39 (38.6) | 29 (38.7) |  |
| Telemedicine | 12 (6.8) | 6 (5.9) | 6 (8.0) |  |
| General practitioner | 83 (47.2) | 47 (46.5) | 36 (48.0) |  |
| Other | 8 (4.5) | 6 (5.9) | 2 (2.7) | 0.859 |
| Nationality |  |  |  |  |
| Not want to answer | 1 (0.6) | 1 (1.0) | 0 (0.0) |  |
| Switzerland | 147 (83.5) | 80 (79.2) | 67 (89.3) |  |
| Germany | 13 (7.4) | 8 (7.9) | 5 (6.7) |  |
| French | 1 (0.6) | 0 (0.0) | 1 (1.3) |  |
| Italy | 3 (1.7) | 2 (2.0) | 1 (1.3) |  |
| Other Europe | 4 (2.3) | 3 (3.0) | 1 (1.3) |  |
| Other | 7 (4.0) | 7 (6.9) | 0 (0.0) | 0.202 |

*$\chi^2$ for categorical variables and Wilcoxon rank sum test for continuous variables; data are total number and percentage if not mentioned otherwise.

to structure the report of our findings, that is, (1) accessibility of the tool, (2) user-friendliness of the tool, (3) utility of the tool as an information source, (4) utility of the tool in allaying fear and anxiety, (5) utility of the tool in decision-making (test or not to test), (6) utility of the tool in reducing onward transmission–cross infection and (7) utility of the tool in reducing health system burden. The qualitative findings are summarised in table 5.

### Theme 1: accessibility of the tool
The accessibility of the tool emerged as very important. Many participants suggested to advertise the tool to make it more accessible as revealed below:

I did not know of the existence of tool (an accidental internet search led the key informant to the tool). Please advertise tool on TV and to Insurance companies. -Key Informant 15

The older people seem willing to embrace technology and were prepared to use it. However, they stated that they needed help with practical application at times as revealed below:

Provide telephone services for the elderly and a contact person, a GP so one can ask questions if unsure. -Key Informant 14

**Table 3** Online forward triage tool use

|  | Total (n=176) (%) |
|---|---|
| **Information searched** | |
| Information on COVID-19 symptoms | 97 (55.1) |
| How to cope with symptoms | 4 (2.3) |
| To know when to consult a doctor | 36 (20.5) |
| To know more on testing criteria | 32 (18.2) |
| To know where to test | 7 (4.0) |
| **Mode of referral** | |
| Referral by family doctor | 9 (5.1) |
| Online search | 113 (64.2) |
| Recommendation by peers | 17 (9.7) |
| Hotline | 2 (1.1) |
| Other | 35 (19.9) |
| **Satisfaction with information** | |
| Helpful | 154 (87.5) |
| Not comprehensive | 17 (9.7) |
| Not clear | 5 (2.8) |

### Theme 2: user-friendliness of the tool

Most participants could not remember the tool immediately due to the timelapse from the tool usage to interview. After being shown the tool once again, the header only, many cited it as having been easy and simple to follow with the language being clear and the length acceptable.

**Table 4** Additional factors

|  | Total (n=176) (%) |
|---|---|
| **Estimated influence of media** | |
| Helpful | 81 (46.0) |
| Confusing | 47 (26.7) |
| No trust in media as source of information | 25 (14.2) |
| Other | 23 (13.1) |
| **Influence of OFTT on fear and anxieties** | |
| Reassured | 73 (41.5) |
| No reassurance | 13 (7.4) |
| Increased fears and anxieties. | 6 (3.4) |
| Not worried before OFTT use | 84 (47.7) |
| **Reasons for following the recommendation (n=149)** | |
| Trust in tool | 60 (40.3) |
| Information congruent with media | 20 (13.4) |
| Comparison with FOPH recommendation | 53 (35.6) |
| Reassurance by others | 7 (4.7) |
| Other | 9 (6.0) |

FOPH, Federal Office of Public Health; OFTT, online forward triage tool.

### Theme 3: utility of the tool as an information source

The novel nature of COVID-19 infection left many scrambling for knowledge of the disease. Many healthcare providers were inundated with phone calls. One participant said the following:

> The tool provided information on symptoms but did not have a list of testing centers. The recommendations said call GP before visit but there was no number to call. -Key Informant 1

> Telemedicine could play a better information spreading role – media spread fear and misinformed people for example mask use vs no mask. -Key Informant 15

### Theme 4: utility of the tool in allaying fear and anxiety

Many participants interviewed reported being reassured after tool use. Others cited being more anxious after tool use due to terminology and language and many suggested that a person, a doctor be available after tool use for closure. Participants revealed the following:

> Wording of tool could be adapted – a friend aged 65, a diabetic, became depressed after using tool and getting the high-risk patient classification. He needed a psychiatrist to cope. Rather ask how are you, do you take any medication, which ones? Mentioning conditions seem to increase anxiety. -Key Informant 17

> I felt discriminated against by tool-differentiate between a health 73-year-old with no chronic illnesses and a 50-year overweight diabetic. -Key Informant 13

### Theme 5: utility of the tool in decision-making process (to test or not to test)

Many participants cited trust in our university hospital (Insel) as one of the main reasons participants followed the recommendations. Some participants revealed the following:

> Insel has a good name and trusted the tool. -Key Informant 16

> Coordination is needed for FOPH and Insel to speak in one voice. -Key Informant 17

Juxtaposed and not necessarily contradicting the quantitative survey, where trust was reported as the main reason for following the recommendations, most of the participants cited shortages of tests, improved symptoms, cost of test, misinformation that the test was painful and fear of a positive result as reasons for not testing. Of utmost importance were GPs who viewed the test request by online tool users as being hysteric. Below is what some participants said:

> I read scientific papers to inform oneself and then decided. -Key Informant 8

> Remember recommendations from an online tool have less weight than recommendations from a doctor – there is no person behind this and so many

**Table 5** Summary of qualitative themes

| Theme | Category | Unit meaning |
|---|---|---|
| Accessibility | Online search<br>Unreachable for some | Appeared but not on the top of google search<br>Advertise tool in future<br>Include telephonic services to reach the elderly<br>Tool buddies |
| Utility as a reliable information source | COVID-19 symptoms<br>Testing info and centres missing | Cough was a main symptom<br>Symptom description such as type of cough and severity of fever etc was not possible<br>Test or do not test decision was arbitrary—how the decision was arrived at was not clear, for example, 95% probability test or 5% probability do not test<br>Information on when to call doctor was not clear, for example, fever above 39°C for 4 days—call doctor<br>List of where to test and contact numbers were missing |
| Utility in decision-making | Followed recommendations<br>Did not follow recommendations | Trust—the university hospital is a trusted institution<br>Fear of a positive result and the resultant consequences<br>Cost of test<br>Test shortage<br>GP refusing patients to test—hysteria |
| Utility in allaying fear and anxiety | Reassured some<br>Person contact<br>Testing<br>Friends and family as a resource<br>Increased anxiety in some | Fear and anxiety allayed after tool use<br>An online tool is still an online tool—recommendations seen as not having a lot of weight<br>A talk with a GP—debriefing after tool use could have put them at ease<br>Testing in itself is reassuring—make test available to all who are anxious<br>Many relied on family and friends to deal with fear—social circle still a major source of support<br>High-risk label unsettled some |
| Utility in reducing health system burden | Many stayed at home | Recommendations followed—stay at home<br>Some called Insurance companies |
| Utility in reducing onward transmission | Call general practitioner (GP) before a visit | Most called GP ahead of visit |
| Systems thinking | Utility of tool is dependent on other health system and societal components<br>Fear of a positive test, rather not know | Participants told by tool to test only to be told that there are no tests (shortages)<br>Fear of a positive test<br>Media misinformation of painful test influenced some not to test—work with media<br>Economic factors like cost of test influenced some not to test<br>A new life-threatening disease in a population is associated with psychosocial and behavioural issues that need to be taken into account |

might have taken the tool and went further to contact own GP- Key Informant 8

I wished to see an algorithm that said something like, "the probability of you having COVID-19 is 75% test or 25% do not test.-Key Informant 5

**Theme 6: utility in reducing the potential for onward transmission– cross infection**

The tool recommended all participants to call the health-care provider ahead of visit and most of them did. A reason some participants might not have called the testing centres ahead of a visit could be that the tool itself did not provide a list of contact numbers—a shortcoming that was rectified in the second-generation OFTT.

**Theme 7: utility of tool in reducing health system burden**

Social distancing, isolation and quarantine were among the recommendations made to reduce the spread of COVID-19. Most of the participants stayed at home. One participant said the following;

I followed recommendations and stayed at home. However, home testing should be provided if people should stay at home. Engage Spitex [organization for outpatient and home-based care in Switzerland] in future pandemics and work with them. -Key Informant 6

**DISCUSSION**

This study quantitatively assessed the effects and confirmed the utility (qualitatively) of a COVID-19 OFTT by exploring patient perspectives. We further elaborate on areas for improvement as well as share lessons learnt for policy-makers. Qualitatively, seven overarching

themes emerged namely (1) accessibility of tool, (2) user-friendliness of tool, (3) utility of tool as an information source, (4) utility of tool in allaying fear and anxiety, (5) utility of tool in decision-making (test or not to test), (6) utility of tool in reducing the potential for onward transmissions (preventing cross infection) and (7) utility of tool in reducing health system burden.

## Accessibility of OFTT

One of the objectives of our OFTT was to provide an easily accessible, reliable and up to date information platform for professionals and the public. The tool was not advertised commercially; hence, it did not appear at the top of the Google search and many participants cited coming across the tool accidentally. Information about the tool was only disseminated via the hospital website and hospital communication to local doctors.

Despite the above-mentioned shortcoming, our findings revealed that the tool was accessible to both genders and all age groups including the elderly. In line with other studies,[21] the elderly seem ready to embrace online tools, contradicting other studies.[10 17] Contradicting our findings, one study revealed that it is the young and highly educated patients who tend to use symptom checkers or OFTTs.[22]

Despite the revealed readiness of the elderly to embrace technology, key informants suggested keeping the use of telephonic services for the elderly as an option in telemedicine. Further supporting these findings, nurse triage lines (telephone) have been proven effective in this COVID-19 pandemic context in the USA and in Canton Vaud, Switzerland.[10 23] Others suggested having a list of tool buddies reachable by phone that links people who have used the tool before and are willing to be contacted by a new user, who might be experiencing challenges in using the OFTT. With regard to reaching the low education and low-income group, additional studies need to be done as those who earned less than SFr4000 were not necessarily lowly educated but PhD and post-doc students, concurring with findings elsewhere.[24]

## User-friendliness of OFTT

Most of the participants could not recall tool, but after showing them tool header only, many cited tools as user-friendly, easy, with a clear language and an acceptable length, concurring with a study that was conducted elsewhere.[25] In support of our findings, online tools have been shown to be risk averse as compared with healthcare professionals and the users have expressed high levels of satisfaction.[22] The optimal amount of time spent filling in OFTT questionnaires nor the optimal number of questions an OFTT should ask in general is still unclear[26] and warrants further studies.

## Utility of OFTT as an information source

Overall, the tool was very useful in providing information on signs and symptoms. Information on where to test (list with contact numbers), how to self-care, when

to contact a GP were cited by some as shortcomings and ought to be included to make the tool comprehensive in future. Information challenges with OFTTs have also been reported elsewhere.[27 28] This finding underlines the need to have an option to talk directly to a GP after OFTT use so as to debrief.

Further information or links to comprehensive and reliable sources with information on how to self-care and when to contact a GP or healthcare centre emerged as gaps that need to be incorporated in COVID-19 OFTTs so as to increase their utility as information sources. The majority of our participants were highly educated, and this segment of the population seems to inform itself, by consulting a variety of scientific sources as well as keeping abreast with the FOPH announcements. In the context of a novel infection, where guidelines change quickly and continuously, the credibility of the tool to the highly educated, could be enhanced by stipulating sources of information and referencing and dating the FOPH criteria informing the tool.

## Utility of OFTT in allaying fear and anxiety

For most of the participants, the tool was effective in allaying their fear and anxiety. Many wished a human presence, a doctor to debrief with after the online tool use as mentioned above. There was, however, a downside for some that felt labelled as being high risk. For this group, the tool had a negative effect and increased their anxiety. Other studies have revealed similar effects.[29 30] This raises the issue of language and terminology use in such tools. Bearing in mind that COVID-19 is a novel condition, not well understood and considered fatal, the impact of a high-risk label should not be underestimated, including discrimination. Concurring with our findings, COVID-19 stigma has been reported elsewhere.[31] Many participants reported fear of a positive test result and the consequences thereof, concurring with findings from elsewhere.[32 33] Further concurring with our findings, lasting psychological consequences that last beyond the COVID-19 infection itself have also been revealed.[31] This raises the question of psychological readiness to deal with such a diagnosis. Emerging studies have reported patients with COVID-19 as having psychiatric-related conditions post infection, further concurring with our study.[34 35]

## Utility of OFTT in facilitating decision-making

The tool was useful in assisting patients in decision-making particularly not to test. Trust in the institution proved pivotal as many followed recommendations simply because they trusted the source of the tool, our university hospital. Studies elsewhere concur with our findings.[36 37] On the other hand, some of those that got the recommendation to test did not do so due to a myriad of reasons as revealed above. In addition, the cost of the test (SFr180 at the time), shortages of tests and fear of a positive result and the resultant

consequences of isolating, stigma etc further influenced decisions not to test. A low income was found not to be a reliable socioeconomic status proxy in our study. Most low-income participants were PhD students and post-docs who cited various reasons for not following recommendations. Many told us how they sought and read scientific evidence to inform themselves and this, rather than the recommendations, guided their decision-making. In line with our findings, salary is not a good proxy for socioeconomic status among online tool users.[24] A shortcoming in this regard was the missing information on how the tool arrived at the recommendation to test or not to test, for example, algorithm used[19] something some key informants wished to know. The issue of safety concerns with regard to specificity of digital tool algorithms has also been reported elsewhere.[38]

## Utility of OFTT in preventing onward transmission–cross infection

The tool proved useful in preventing cross infection concurring with findings elsewhere.[19] Most participants who were told to stay at home did so, reducing mobility and exposure. Most of the participants called the GP practice ahead of time. That gave the GP practices time to ensure that the suspect patient did not mix with other patients, thereby reducing the potential for onward transmission (cross infection).[19]

## Utility of OFTT in reducing health system burden

Our primary hypothesis was that such an OFTT reduces the health system burden. Most of the participants who used the tool would have called their GP or visited the hospital. OFTT use effectively kept these worried participants at home and out of the doctors' offices and hospitals, effectively reducing the health system burden. Contradicting our findings, research from elsewhere has produced inconclusive and sometimes contradicting evidence.[28 39] Further studies in different contexts are therefore called for. Further contradicting our findings, another study reported that symptom checkers' triage capabilities are not greater than that of an average lay person.[40] In fact, the convenience of telemedicine has also been associated with increased utilisation of services, increasing work load and healthcare spending.[41]

## Recommendations and lessons learned

Our study demonstrated the effects and utility of a COVID-19 OFTT. The assessment of an OFTT is important but not without challenges. Below are some of the lessons worth sharing with both healthcare providers and policymakers as subsequent waves sweep across Europe:

► Most of the participants had challenges remembering the tool. Immediate feedback, for example, in 1 min, please rate this tool, or three open questions; please tell us how useful this tool was with regard to (a) accessibility of tool, (b) utility of tool as an information source, (c) utility of tool in facilitating your decision-making could be more effective. Data protection concerns and the need to keep barriers to use as low as possible could stand in the way of this approach.

► The tool simply instructed patients to test or not to test, an arbitrary decision, without shedding light on how the decision was made. Patients wish to see an algorithm that says something like, 'the probability of you having COVID-19 is 75% test or 25% do not test'.

► Many participants said, 'bear in mind that online tool recommendations have less weight than recommendations from a GP'. Additional caution is needed in language and terminology use as some patients who felt labelled by tool as high risk, had negative outcomes. Ensuring access to a doctor to debrief with after such tool use is advisable. Retired doctors who are still willing to make a contribution to the society could play such a role.

► Many participants found the tool by accident; hence, it is advisable to advertise tool on social media platforms, billboards, TV, radio and could make it appear at the top of Google search. In addition, taking the tool to the people, for example, through road shows could be a useful strategy to reach the old people— if they do not come to the tool, take the tool to the people.

► Many participants compared the tool recommendations with what the FOPH recommended at the time. Having a tool link on FOPH website that stipulates and references the FOPH criteria informing the tool could increase trust in tool and acceptability. Coordination among FOPH, university hospitals and other medical professional bodies is recommended to further enhance trust in the tool.

► Many elderly people are willing to embrace telemedicine, but challenges persist. Telephone and voice-activated system for the older population or call centres to serve this group are still needed (taking heed of unreachable and unanswered calls) during this transitional phase.

► Most participants found media confusing—telemedicine could play a better information spreading role, sifting through the noise and offering scientific based recommendations. For many, the media spread fear and misinformed people in many instances.

► The OFTT lacked information on where to test (contact list of testing centres), how to self-care, how to manage symptoms and when to contact a doctor— addressing these shortcomings could improve the utility of OFTTs. Our results underline the importance of not offering a telehealth tool as a stand-alone product, but to integrate it into an overall concept with links to credible reliable sources.

► Systems thinking refers to the ability to see interconnectedness in a system with a dysfunction in one part affecting other parts and consequently outcomes. Our study revealed the reasons patients did not follow the recommendation to test, as multipronged.

Attention has to be paid to supply chain issues, as test shortages affected outcomes. The cost of a test and the fear of a positive result additionally emerged as hindrances to testing. This calls for systems thinking. Noteworthy, is the reaction of GPs who labelled OFTT users who asked for a COVID-19 test as hysteric. This does not only reveal that the pandemic caught everyone by surprise but also demonstrates the need to involve, collaborate with and win the local healthcare providers policy implementers, such as GPs and Spitex (home-based nursing), to enhance tool utility as well as ensure positive outcomes

► One key informant suggested having patients who had recovered from COVID-19 act as champions to share their illness experience, and motivate the public to take preventive measures and take the disease seriously—an approach that was also effective in HIV prevention and coping strategies.

## Strengths and limitations

Many online tools have been developed during the COVID-19 pandemic. The effects and utility of these tools, however, have not been assessed. Coronatest.ch was one of the first COVID-19 OFTTs in Switzerland. Our study could become the baseline for studies that assess the effects and utility of such online tools. The identified themes namely (1) accessibility of tool, (2) user-friendliness of tool, (3) utility of tool as an information source, (4) utility of tool in allaying fear and anxiety, (5) utility of tool in decision-making (test or not to test) and (6) utility of tool in reducing onward transmission–cross infection, (7) utility of tool in reducing health system burden, could serve as a framework for assessing OFTT utility (follow-up paper). The mixed method sequential explanatory design gave us a better understanding of OFTTs, their effects measured quantitatively and utility explained with the aid of qualitative findings. We did not simply report the effects but could also explain why the results were that way, generating a holistic picture of the phenomenon.

The selection of the participants in our study carries the risk of a selection bias. Perspectives of those that do not use online tools are missing and should be explored in further studies. In addition, only a limited number of OFTT users took part in our study. This selection bias cannot, to the best of our knowledge, be prevented due to data protection regulations, which impose a voluntary participation and prohibit a technically possible automatic tracking of participants. Another way to avoid this possible selection bias would be to make the use of such a tool conditional on participation in the study. We have deliberately decided against this procedure for ethical reasons, in order to make our OFTT accessible to as many users as possible and to keep barriers as low as possible. In addition, mandatory entry of personal data in OFTT for study purposes would also discourage individuals from using the tool and thus trigger a new bias. Our comparison of overlapping questions between the OFTT and the follow-up survey can at least help to estimate the similarities within the two groups. For both questions, the percentages are comparable and can help in estimating the similarity of the groups.

Another limit of our study is the relatively long duration between the use of tool and the qualitative interviews. This could have introduced a certain degree of recall bias. As with all online tools, we cannot confirm the accuracy of the data entered. In particular, we cannot say for sure whether the OFTT users used the tool to assess own symptoms or for other reasons, such as curiosity, fear or uncertainty about how to deal with the novel infection. Likewise, multiple use, trial runs or use of tool by a healthcare worker on behalf of patients, relatives and friends are all possible. Socioeconomic status might have introduced a selection bias in our study, since most of the participants had a higher education. Income emerged not to be a good proxy for assessing socioeconomic status. Other instruments, apart from income, are therefore needed to assess socioeconomic status. Additionally, an on online assessment cannot fully replace a PCR test as some asymptomatic people might be positive and those with COVID-19-specific symptoms might be suffering from a different disease.[5] In our mind, the data still sheds light on the effects and utility of such an online tool and the recommendations given could guide other OFTT developers as the third wave sweeps across Europe. As the study was conducted with a specific OFTT, transferability of our results to other OFTTs is not necessarily a given. Given the limited evidence on the use of OFTTs, the results, in particular the qualitative component of the study, could be of value to other OFTT developers, with particular regards to utility and accessibility issues. Further studies with other OFTTs outside the COVID-19 context are recommended so as to increase transferability and improve the utility of OFTTs in the current third wave, future pandemics and other healthcare settings.

## CONCLUSION

OFTT use has increased greatly during this pandemic. The effects and utility of such tools, however, have not been widely assessed. That makes our study, one of the firsts, in assessing effects and utility of a COVID-19 OFTT. Our study revealed that an OFTT does not only reduce the health system burden but can also serve as an information source, reduce anxiety and fear, reduces potential for onward transmission and facilitate decision-making.

**Acknowledgements** Emergency telemedicine at University of Bern, Switzerland, is supported by an endowed professorship by the Touring Club Switzerland.

**Contributors** Study design and idea: JM, AM, RS, TCS, MM, MER, PJ and WEH. Data extraction and preparation: RS, MM and AM. Qualitative interviews: JM, RS and AM. Statistical analysis: MM. Qualitative analysis: JM. Writing of first draft: JM, AM and MM. Revision of the final draft and final approval: all authors. Supervision:

TCS and WEH. Project administration: TCS and WEH. TCS is the senior author and guarantor of the reasearch project.

**Funding** This manuscript is partially founded by the Swiss National Science Foundation (Project ID: 196615).

**Competing interests** TCS holds the endowed professorship for emergency telemedicine at University of Bern, Switzerland.

**Patient and public involvement** Patients and/or the public were not involved in the design, or conduct, or reporting, or dissemination plans of this research.

**Patient consent for publication** Consent obtained directly from patient(s)

**Ethics approval** The local ethics committee of the Canton of Bern, Switzerland, deemed this project a quality evaluation study and waived the need for full ethical review (Req-2020-00289) on 23 March 2020.

**Provenance and peer review** Not commissioned; externally peer reviewed.

**Data availability statement** No data are available. Due to the nature of the study (online forward triage tool, OFTT) participants did not agree for their data to be shared publicly. The data to support findings are available. Please contact corresponding author JM. Any requests can be sent to the corresponding author.

**ORCID iDs**
Janet Michel http://orcid.org/0000-0002-8412-219X
Raphael Stuber http://orcid.org/0000-0002-8213-4285
Philipp Jent http://orcid.org/0000-0002-1838-9208
Wolf E Hautz http://orcid.org/0000-0002-2445-984X
Thomas C Sauter http://orcid.org/0000-0002-6646-5789

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
