## [Reviewer comments · BMJ Open]

ARTICLE DETAILS

TITLE (PROVISIONAL)	Effects and utility of an online forward triage tool during the SARS-CoV-2 pandemic: a mixed method study and patient perspectives, Switzerland
AUTHORS	Michel, Janet; Mettler, Annette; Stuber, Raphael; Müller, Martin; Ricklin, Meret; Jent, Philipp; Hautz, Wolf; Sauter, Thomas

VERSION 1 – REVIEW

REVIEWER	Pulia, Michael University of Wisconsin Madison School of Medicine and Public Health, Emergency Medicine
REVIEW RETURNED	13-Feb-2022

GENERAL COMMENTS	The authors present the results of a mixed methods study examine the effect and utility of an online triage tool for patients with COVID-19 related concerns. The tool was active in the early months of the pandemic and the specific recommendations of the tool evolved based on pandemic epidemiology and testing supply dynamics. Users of the tool were invited to participate in a follow-up survey that sought to capture information about why they used the tool, adherence to the guidance, and how it addressed their concerns. The provided quantitative and qualitative analyses were conducted with appropriate and clearly reported methodology. The provision of the survey instrument and qualitative interview guide for review was appreciated. The major strength of this work is the synergistic presentation of quantitative survey results that sequentially informed the qualitative analysis. Although the specifics of online COVID triage tools are anticipated to evolve at each stage of the pandemic, the information gathered in this study is of a general scope and the end user input will be incredible useful in the design of future iterations. Below are several specific minor concerns/comments for consideration. -The relatively long duration of time between use of the tool and the qualitative interviews is a limitation as it relies heavily on recall which is prone to bias-Additional information on the exact qualitative analysis approach and underlying theory would be useful (e.g. ground theory, directed content, etc)-Unclear why the demographic Table 2 includes a statistical comparison of group demographics by sex. This seems to be an arbitrary comparison point as compared to other potential categories such as age or adherence to tool recommendation.-Theme 4 relates to the tools role in allaying fear and anxiety yet the two exemplar quotes are highlighting negative effects of the tool on this
--

REVIEWER	Salman, Omar H Al Iraqia University, Network Department
REVIEW RETURNED	14-Feb-2022

GENERAL COMMENTS	the research topic is interesting. The structure of the article is well presented. However, the authors need to add one figure in the introduction section to show how does the OFTL works? as well as the authors need to highlight the performance of the OFTL with other triage techniques that are currently used in emergency department.
--

VERSION 1 – AUTHOR RESPONSE

Reviewer: 1: Dr. Michael Pulia, University of Wisconsin Madison School of Medicine and Public Health

Reviewer comment 1: The authors present the results of a mixed methods study examine the effect and utility of an online triage tool for patients with COVID-19 related concerns. The tool was active in the early months of the pandemic and the specific recommendations of the tool evolved based on pandemic epidemiology and testing supply dynamics. Users of the tool were invited to participate in a follow-up survey that sought to capture information about why they used the tool, adherence to the guidance, and how it addressed their concerns. The provided quantitative and qualitative analyses were conducted with appropriate and clearly reported methodology. The provision of the survey instrument and qualitative interview guide for review was appreciated. The major strength of this work is the synergistic presentation of quantitative survey results that sequentially informed the qualitative analysis. Although the specifics of online COVID triage tools are anticipated to evolve at each stage of the pandemic, the information gathered in this study is of a general scope and the end user input will be incredible useful in the design of future iterations.

*Author response

We thank the reviewers for the encouraging comments. We fully agree with the reviewers that end user input is of utmost importance in the design of future iterations. We have come up with such a proposal and have submitted an application for a grant. Depending on funding, we hope to involve both end users and health care providers in the design and evaluation of our telehealth interventions in future.

Reviewer comment 2: Below are several specific minor concerns/comments for consideration. The relatively long duration of time between use of the tool and the qualitative interviews is a limitation as it relies heavily on recall which is prone to bias.

*Author response

We agree with the reviewers that the long duration time between use of the tool and interviews is one of the limitations of our study. We thank the reviewers for this point. We have now incorporated a statement in our study strengths and limitations. It now reads as follows;

Line 144-145

Another limit of our study is the relatively long duration between the use of tool and the qualitative interviews. This could have introduced a certain degree of recall bias.

Reviewer comment 3: Additional information on the exact qualitative analysis approach and underlying theory would be useful (e.g., ground theory, directed content, etc)

*Author response

Thank you very much for asking such a pertinent question. We have now incorporated a response into the manuscript. Lines 358-363 now read as follows;

Audio recordings were transcribed, analysed and triangulated with quantitative data results. Qualitative narratives were obtained to explain quantitative results as well as to explore utility of OFTT to patients as well as elicit recommendations to make online tools more useful and inclusive. A grounded theory approach was utilized. Concepts were identified from collected data and compared iteratively. These concepts were grouped into categories and culminated into the identified themes.

Reviewer comment 4: Unclear why the demographic Table 2 includes a statistical comparison of group demographics by sex. This seems to be an arbitrary comparison point as compared to other potential categories such as age or adherence to tool recommendation.

*Author response

We thank the reviewer for the comments. One of our interests was to explore tool use along gender lines. A related but separate study by our team had identified such a pattern and we used this as a departure point, hence we compared tool use by sex. The reference is below;

Annette C. Mettler, Livio Piazza, Janet Michel, Martin Müller, Aristomenis K. Exadaktylos, Wolf E. Hautz, Thomas C. Sauter. Use of telehealth and outcomes before a visit to the emergency department: a cross-sectional study on walk-in patients in Switzerland. *Swiss Med Wkly.* 2021;151: w20543

Theme 4 relates to the tool's role in allaying fear and anxiety yet the two exemplar quotes are highlighting negative effects of the tool on this.

*Author response

We agree with the reviewer. Most of the participants revealed that the fear of having the disease was allayed when they got the recommendation do not test. The question, "Do you have a high-risk factor e.g., aged above 65 or having a chronic disease," was cited as unsettling for some. We reported this as it is a central tenet of qualitative research to have a balanced presentation of findings, both positive and negative (1–3). The findings presented show the limited utility of the tool in allaying fear and anxiety since not all fears can be allayed. We also share this as a lesson learned. Lines 653-655 read as follows;

Additional caution is needed in language and terminology use as some patients that felt labelled by tool as high risk, had negative outcomes. Ensuring access to a doctor to debrief with after such tool use is advisable.

Reviewer 2: Dr. Omar H Salman, Al Iraqia University

Reviewer Comment 1: The research topic is interesting. The structure of the article is well presented. However, the authors need to add one figure in the introduction section to show how does the OFTL works? as well as the authors need to highlight the performance of the OFTL with other triage techniques that are currently used in emergency department. How does the OFTT work as compared

to other triage techniques

*Author response

Thank you for the comment and the request for an explanation. We have now done that and it now reads as follows; Lines 253-264

Details on the structure of the OFTT as well as screen shot are published in a separate quantitative paper(4). The Federal office of public health (FOPH) national COVID-19 Swiss testing criteria were transferred into a digital decision tree and adjusted promptly after the criteria were adapted by the FOPH. During the first phase of the pandemic, the recommendations for testing or not testing were mainly based on contact with an infected person or a visit to a risk area and were then changed during the course of the pandemic to a testing regime based on risk groups (healthcare professionals, patients >65 years and patients with pre-existing conditions). With the general availability of the tests, the test recommendations were extended to all symptomatic patients and our OFTT became obsolete. Unlike other triage techniques performed on emergency patients, the aim of the OFTT was not to make a COVID-19 diagnosis, assess the risk of severe COVID-19 progression or recommend treatment. See Fig 2 below;

Fig 2 OFTT triage

References

1. Braun, V., Clarke, V. Using thematic analysis in psychology. *Qual Res Psychol.* 2006;(3):77–101.
2. Ivankova NV, Creswell JW, Stick SL. Using Mixed-Methods Sequential Explanatory Design: From Theory to Practice. 2006; Verfügbar unter: https://pdfs.semanticscholar.org/363c/fe5efa01349c3685a023950ffa552ae824bf.pdf?_ga=2.201592529.1389452148.1584978635-451654905.1584978635
3. Baxter J, Jack S. Qualitative Case Study Methodology: Study Design and Implementation for Novice Researchers. *Qual Rep.* 2008;
4. Hautz WE, Exadaktylos A, Sauter TC. Online forward triage during the COVID-19 outbreak. *Emerg Med J EMJ.* Februar 2021;38(2):106–8.